# Mode-Aware Continual Learning for Conditional Generative Adversarial Networks

## Abstract

The main challenge in continual learning for generative models is to effectively learn new target modes with limited samples while preserving previously learned ones. To this end, we introduce a new continual learning approach for generative modeling in conjunction with a mode-affinity score specifically designed for conditional generative adversarial networks. First, the generator produces samples of existing modes for subsequent replay. The discriminator is then used to compute the mode similarity measure, which identifies a set of closest existing modes to the target. Subsequently, a label for the target mode is generated and given as a weighted average of the labels within this set. We extend the continual learning model by training it on the target data with the newly-generated label, while performing memory replay to mitigate the risk of catastrophic forgetting. Experimental results on benchmark datasets demonstrate the gains of our approach over the state-of-the-art methods, even when using fewer training samples.

## 1 Introduction

Artificial intelligence (AI) for generative tasks has made significant progress in recent years, and we have seen remarkable applications, such as ChatGPT (OpenAI, 2021), DALL-E (Vaswani et al., 2021), and deepfake (Westerlund, 2019). Nevertheless, most of these methods Wang et al. (2018); Varshney et al. (2021); Zhai et al. (2019); Le et al. (2020); Seff et al. (2017) lack the ability to learn continuously, which remains a challenging problem in developing AI models that can match human's continuous learning capabilities. This challenge is particularly difficult when the target data is limited or scarce (Varshney et al., 2021; Zhai et al., 2019; Seff et al., 2017). In this scenario, the goal is to learn to generate new images using an extensive model that is trained on all previous tasks. Most continual learning methods focus on preventing the models from forgetting the existing tasks. However, many learning restrictions are often enforced on learning new tasks, leading to poor performance. To address this issue, relevant knowledge from previously learned tasks is identified and harnessed to efficiently learn the new tasks. Various knowledge transfer approaches have been introduced, resulting in significant breakthroughs in many applications, including natural language processing (Vaswani et al., 2017; Devlin et al., 2018; Howard & Ruder, 2018; Le et al., 2023; Brown et al., 2020), and image classification (Elaraby et al., 2022; Guo et al., 2019; Ge & Yu, 2017; Le et al., 2022b; Cui et al., 2018; Azizi et al., 2021). These techniques enable models to leverage past experiences, such as trained models, and hyper-parameters to improve their performance on the new task, emulating how humans learn and adapt to new challenges (e.g., learning to ride a motorcycle is less challenging for someone who already knows how to ride a bicycle). It is also essential to identify the most relevant task for knowledge transfer when dealing with multiple learned tasks. Irrelevant knowledge can be harmful when learning new tasks (Le et al., 2022b; Standley et al., 2020b), resulting in flawed conclusions (e.g., misclassifying dolphins as fish instead of mammals could lead to misconceptions about their reproduction).

In this paper, we propose a *Discriminator-based Mode Affinity Score* (dMAS) to evaluate the similarity between generative tasks and present a new continual learning approach for the conditional generative adversarial network (cGAN) (Mirza & Osindero, 2014). Our approach allows for seamless and efficient integration of new tasks' knowledge by identifying and utilizing suitable information from previously learned modes. Here, each mode corresponds to a generative task. Our framework first evaluates the similarity between the existing modes and the new task using dMAS. It enables the identification of the most relevant modes whose knowledge can be leveraged for quick learning of the

target task while preserving the knowledge of the existing modes. To this end, we add an additional mode to the generative model to represent the target task. This mode is assigned a class label derived from the labels of the relevant modes and the computed distances. Moreover, we incorporate memory replay (Robins, 1995; Chenshen et al., 2018) to prevent catastrophic forgetting.

Extensive experiments are conducted on the MNIST (LeCun et al., 2010), CIFAR-10 (Krizhevsky et al., 2009), CIFAR-100 (Krizhevsky et al., 2009), and Oxford Flower (Nilsback & Zisserman, 2008) datasets to validate the efficacy of our proposed approach. We empirically demonstrate the stability and robustness of dMAS, showing that it is invariant to the model initialization. Next, we apply this measure to the continual learning scenario. Here, dMAS helps significantly reduce the required data samples, and effectively utilize knowledge from the learned modes to learn new tasks. We achieve competitive results compared with baselines and the state-of-the-art approaches, including individual learning (Mirza & Osindero, 2014), sequential fine-tuning (Wang et al., 2018), multi-task learning (Standley et al., 2020b), EWC-GAN (Seff et al., 2017), Lifelong-GAN (Zhai et al., 2019), and CAM-GAN (Varshney et al., 2021). The contributions of our paper are summarized below:

- We propose a new discriminator-based mode-affinity measure (dMAS), to quantify the similarity between modes in conditional generative adversarial networks.
- We provide theoretical analysis (i.e., Theorem 1) and empirical evaluation to demonstrate the robustness and stability of dMAS.
- We present a new mode-aware continual learning framework using dMAS for cGAN that adds the target mode to the model via the weighted label from the relevant learned modes.

## 2 RELATED WORKS

Continual learning involves the problem of learning a new task while avoiding catastrophic forgetting (Kirkpatrick et al., 2017; McCloskey & Cohen, 1989; Carpenter & Grossberg, 1987). It has been extensively studied in image classification (Kirkpatrick et al., 2017; Achille et al., 2018; Rebuffi et al., 2017; Verma et al., 2021; Zenke et al., 2017; Wu et al., 2018; Singh et al., 2020; Rajasegaran et al., 2020). In image generation, previous works have addressed continual learning for a small number of tasks or modes in GANs (Mirza & Osindero, 2014). These approaches, such as memory replay (Wu et al., 2018), have been proposed to prevent catastrophic forgetting (Zhai et al., 2019; Cong et al., 2020; Rios & Itti, 2018). However, as the number of modes increases, network expansion (Yoon et al., 2017; Xu & Zhu, 2018; Zhai et al., 2020; Mallya & Lazebnik, 2018; Masana et al., 2020; Rajasegaran et al., 2019) becomes necessary to efficiently learn new modes while retaining previously learned ones. Nevertheless, the excessive increase in the number of parameters remains a major concern.

The concept of task similarity has been widely investigated in transfer learning, which assumes that similar tasks share some common knowledge that can be transferred from one to another. However, existing approaches in transfer learning (Kirkpatrick et al., 2017; Silver & Bennett, 2008; Finn et al., 2016; Mihalkova et al., 2007; Niculescu-Mizil & Caruana, 2007; Luo et al., 2017; Razavian et al., 2014; Pan & Yang, 2010; Zamir et al., 2018; Chen et al., 2018) mostly focus on sharing the model weights from the learned tasks to the new task without explicitly identifying the closest tasks. In recent years, several works (Le et al., 2022b; Zamir et al., 2018; Le et al., 2021b; 2022a; 2021a; Aloui et al., 2023; Pal & Balasubramanian, 2019; Dwivedi & Roig., 2019; Achille et al., 2019; Wang et al., 2019; Standley et al., 2020a) have investigated the relationship between image classification tasks and applied relevant knowledge to improve overall performance. However, for the image generation tasks, the common approaches to quantify the similarity between tasks or modes are using common image evaluation metrics, such as Fréchet Inception Distance (FID) (Heusel et al., 2017) and Inception Score (IS) (Salimans et al., 2016). While these metrics can provide meaningful similarity measures between two distributions of images, they do not capture the state of the GAN model and therefore may not be suitable for transfer learning and continual learning. For example, a GAN model trained to generate images for one task may not be useful for another task because this model is not well-trained, even if the images for both tasks are visually similar. In continual learning for image generation (Wang et al., 2018; Varshney et al., 2021; Zhai et al., 2019; Seff et al., 2017), however, mode-affinity has not been explicitly considered. Although some prior works (Zhai et al., 2019; Seff et al., 2017) have explored fine-tuning cGAN models (e.g., WGAN (Arjovsky et al., 2017), BicycleGAN (Zhu et al., 2017)) with regularization techniques, such as Elastic Weight Consolidation (EWC) (Kirkpatrick et al., 2017), or the Knowledge Distillation (Hinton et al., 2015), they did not focus on measuring mode similarity

or selecting the closest modes for knowledge transfer. Other approaches use different assumptions such as global parameters for all modes and individual parameters for particular modes (Varshney et al., 2021). Their proposed task distances also require a well-trained target generator, making them unsuitable for real-world continual learning scenarios.

# 3 PROPOSED APPROACH

## 3.1 MODE AFFINITY SCORE

We consider a conditional generative adversarial network (cGAN) that is trained on a set $S$ of source generative tasks, where each task represents a distinct class of data. The cGAN consists of two key components: the generator $\mathcal{G}$ and the discriminator $\mathcal{D}$. Each source generative task $a \in S$, which is characterized by data $X_a$ and its labels $y_a$, corresponds to a specific *mode* in the well-trained generator $\mathcal{G}$. Let $X_b$ denote the incoming target data. Here, we propose a new mode-affinity measure, called Discriminator-based Mode Affinity Score (dMAS), to showcase the complexity involved in transferring knowledge between different modes in cGAN. This measure involves computing the expectation of Hessian matrices computed from the discriminator's loss function. To calculate the dMAS, we begin by feeding the source data $X_a$ into the discriminator $\mathcal{D}$ to compute the corresponding loss. By taking the second-order derivative of the discriminator's loss with respect to the input, we obtain the source Hessian matrix. Similarly, we repeat this process using the target data $X_b$ as input to the discriminator, resulting in the target Hessian matrix. These matrices offer valuable insights into the significance of the model's parameters concerning the desired data distribution. The dMAS is defined as the Fréchet distance between these Hessian matrices.

**Definition 1** (Discriminator-based Mode Affinity Score). *Consider a well-trained cGAN with discriminator $\mathcal{D}$ and the generator $\mathcal{G}$ that has $S$ learned modes. For the source mode $a \in S$, let $X_a$ denote the real data, and $\tilde{X}_a$ be the generated data from mode $a$ of the generator $\mathcal{G}$. Given $X_b$ is the target real data, $H_a, H_b$ denote the expectation of the Hessian matrices derived from the loss function of the discriminator $\mathcal{D}$ using $\{X_a, \tilde{X}_a\}$ and $\{X_b, \tilde{X}_a\}$, respectively. The distance from the source mode $a$ to the target $b$ is defined to be:*

$$s[a, b] := \frac{1}{\sqrt{2}} \boldsymbol{trace} \left( H_a + H_b - 2H_a^{1/2} H_b^{1/2} \right)^{1/2}. \tag{1}$$

To simplify Equation (1), we approximate the Hessian matrices with their normalized diagonals as computing the full Hessian matrices in the large parameter space of neural networks can be computationally expensive. Hence, dMAS can be expressed as follows:

$$s[a, b] = \frac{1}{\sqrt{2}} \left\| H_a^{1/2} - H_b^{1/2} \right\|_F \tag{2}$$

The procedure to compute dMAS is outlined in function `dMAS()` in Algorithm 1. Our metric spans a range from 0 to 1, where 0 signifies a perfect match, while 1 indicates complete dissimilarity. It is important to note that dMAS exhibits an asymmetric nature, reflecting the inherent ease of knowledge transfer from a complex model to a simpler one, as opposed to the reverse process.

## 3.2 COMPARISON WITH FID

In contrast to the statistical biases observed in metrics such as IS (Salimans et al., 2016) and FID (Heusel et al., 2017) (Chong & Forsyth, 2020), dMAS is purposefully crafted to cater to our specific scenario of interest. It takes into account the state of the cGAN model, encompassing both the discriminator and the generator. This sets it apart from FID, which uses the GoogleNet Inception model to measure the Wasserstein distance to the ground truth distribution. Consequently, it falls short in evaluating the quality of generators and discriminators. Instead of assessing the similarity between Gaussian-like distributions, our proposed dMAS quantifies the Fisher Information distance between between the model weights. Thus, it accurately reflects the current states of the source models. Furthermore, FID has exhibited occasional inconsistency with human judgment, leading to suboptimal knowledge transfer performance (Liu et al., 2018; Wang et al., 2018). In contrast, our measure aligns more closely with human intuition and consistently demonstrates its reliability. It is important to emphasize that dMAS is not limited to the analysis of image data samples; it can be effectively applied to a wide range of data types, including text and multi-modal datasets.

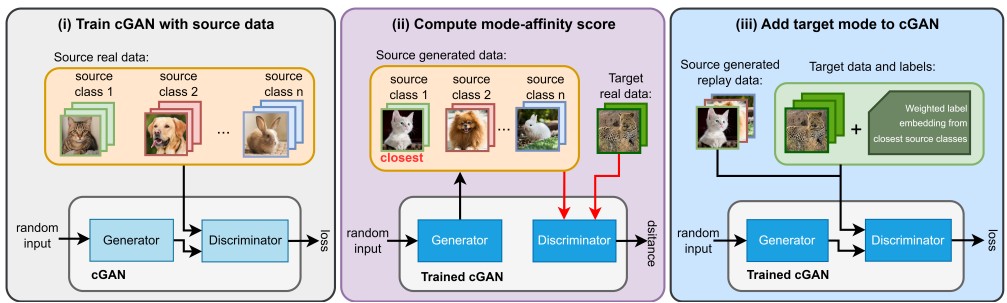

Figure 1: The overview of mode-aware continual learning framework for the conditional Generative Adversarial Network: (i) Representing source data classes using cGAN, (ii) Computing the mode-affinity from each source mode to the target, (iii) Fine-tuning the generative model using the target data and the weighted label embedding from relevant modes for continual learning.

## 3.3 MODE-AWARE CONTINUAL LEARNING FRAMEWORK

We utilize the discriminator-based mode affinity score (dMAS) to continual learning for image generation. The goal is to train a lifelong learning cGAN model to learn new modes while avoiding catastrophic forgetting of existing modes. Consider a scenario where each generative task represents a distinct class of data. At time $t$, the cGAN model has $S$ modes corresponding to $S$ learned tasks. Here, we introduce a *mode-aware continual learning* framework that allows the model to add a new mode while retaining knowledge from previous modes. We begin by embedding the numeric label of each data sample in cGAN, using an embedding layer in both the generator $\mathcal{G}$ and the discriminator $\mathcal{D}$ models. We then modify cGAN to enable it to take a linear combination of label embeddings for the target data. These label embeddings correspond to the most relevant modes, and the weights for these embedding features are associated with the computed dMAS from the related modes to the target. This enables cGAN to add a new target mode while maintaining all existing modes. Let $\mathrm{emb}()$ denote the embedding layers in the generator $\mathcal{G}$ and the discriminator $\mathcal{D}$, and $C$ be the set of the relevant modes, $C = \{i_1^*, i_2^*, \ldots, i_n^*\}$. The computed mode-affinity scores from these modes to the target are denoted as $s_i^*$. Let $\sum s_i^*$ denote the total distance from all the relevant modes to the target. The label embedding for the target data samples is described as follows:

$$\mathrm{emb}(y_{target}) = \frac{s_{i_1^*}}{\sum s_i^*}\mathrm{emb}(y_{train_{i_1^*}}) + \frac{s_{i_2^*}}{\sum s_i^*}\mathrm{emb}(y_{train_{i_2^*}}) + \ldots + \frac{s_{i_n^*}}{\sum s_i^*}\mathrm{emb}(y_{train_{i_n^*}}) \quad (3)$$

To add the new target mode without forgetting the existing learned modes, we use the target data with the above label embedding to fine-tune the cGAN model. Additionally, we utilize memory replay (Robins, 1995; Chenshen et al., 2018) to prevent catastrophic forgetting. Particularly, samples generated from relevant existing modes are used to fine-tune cGAN. The overview of the proposed approach is illustrated in Figure 1. During each iteration, training with the target data and replaying relevant existing modes are jointly implemented using an alternative optimization process. The detail of the framework is provided in Algorithm 1. By applying the closest modes' labels to the target data samples in the embedding space, we can precisely update part of cGAN without sacrificing the generation performance of other existing modes. Overall, utilizing knowledge from past experience helps enhance the performance of cGAN in learning new modes while reducing the amount of the required training data samples. Next, we provide a theoretical analysis of our proposed method.

**Theorem 1.** *Let $\theta$ be the model's parameters and $X_a, X_b$ be the source and target data, with the density functions $p_a, p_b$, respectively. Assume the loss functions $L_a(\theta) = \mathbb{E}[l(X_a; \theta)]$ and $L_b(\theta) = \mathbb{E}[l(X_b; \theta)]$ are strictly convex and have distinct global minima. Let $X_n$ be the mixture of $X_a$ and $X_b$, described by $p_n = \alpha p_a + (1 - \alpha)p_b$, where $\alpha \in (0, 1)$. The corresponding loss function is $L_n(\theta) = \mathbb{E}[l(X_n; \theta)]$. Under these assumptions, it follows that $\theta^* = \arg\min_\theta L_n(\theta)$ satisfies:*

$$L_a(\theta^*) > \min_\theta L_a(\theta) \quad (4)$$

In the above theorem, the introduction of a new mode through mode injection inherently involves a trade-off between the mode-adding ability and the potential performance loss compared to the original model. In essence, when incorporating a new mode, the performance of existing modes cannot be improved. The detailed proof of Theorem 1 is provided in Appendix B.

---

**Algorithm 1:** Mode-Aware Continual Learning for Conditional Generative Adversarial Networks

**Data:** Source data: $(X_{train}, y_{train})$, Target data: $X_{target}$
**Input:** The generator $\mathcal{G}$ and discriminator $\mathcal{D}$ of cGAN
**Output:** Continual learning generator $\mathcal{G}_{\bar{\theta}}$ and discriminator $\mathcal{D}_{\bar{\theta}}$

1 **Function** dMAS $(X_a, y_a, X_b, \mathcal{G}, \mathcal{D})$:

2     Generate data $\tilde{X}_a$ of class label $y_a$ using the generator $\mathcal{G}$

3     Compute $H_a$ from the loss of discriminator $\mathcal{D}$ using $\{X_a, \tilde{X}_a\}$

4     Compute $H_b$ from the loss of discriminator $\mathcal{D}$ using $\{X_b, \tilde{X}_a\}$

5     **return** $s[a, b] = \dfrac{1}{\sqrt{2}} \left\| H_a^{1/2} - H_b^{1/2} \right\|_F$

6 **Function** Main:

7     Train $(\mathcal{G}_\theta, \mathcal{D}_\theta)$ with $X_{train}, y_{train}$            ▷ Pre-train cGAN model

8     Construct S source modes, each from a data class in $y_{train}$

9     **for** $i = 1, 2, \ldots, S$ **do**

10         $s_i = $ dMAS$(X_{train_i}, y_{train_i}, X_{target}, \mathcal{G}_\theta, \mathcal{D}_\theta)$      ▷ Find the closest modes

11     **return** closest mode(s): $i^* = \underset{i}{\operatorname{argmin}}\, s_i$

12     Generate data $X_{i^*}$ of label $i^*$ from closest mode(s)      ▷ Fine-tune for continual learning

13     Define the target label embedding as a linear combination of the label embeddings of the closest modes, where the weights corresponding to $s_{i^*}$ as follows:
$$\operatorname{emb}(y_{target}) = \frac{s_{i_1^*}}{\sum s_i^*} \operatorname{emb}(y_{train_{i_1^*}}) + \ldots + \frac{s_{i_n^*}}{\sum s_i^*} \operatorname{emb}(y_{train_{i_n^*}})$$

14     **while** $\theta$ *not converged* **do**

15         Update $\mathcal{G}_\theta, \mathcal{D}_\theta$ using real data $X_{target}$ and label embedding $\operatorname{emb}(y_{target})$

16         Replay $\mathcal{G}_\theta, \mathcal{D}_\theta$ with generated data $X_{i^*}$ and label embedding $\operatorname{emb}(y_{train_{i^*}})$

17     **return** $\mathcal{G}_{\bar{\theta}}, \mathcal{D}_{\bar{\theta}}$

---

## 4 EXPERIMENTAL STUDY

Our experiments aim to evaluate the effectiveness of the proposed mode-affinity measure in the continual learning framework, as well as the consistency of the discriminator-based mode affinity score for cGAN. We consider a scenario where each generative task corresponds to a single data class in the MNIST (LeCun et al., 2010), CIFAR-10 (Krizhevsky et al., 2009), CIFAR-100 (Krizhevsky et al., 2009), and Oxford Flower (Nilsback & Zisserman, 2008) datasets. Here, we compare the proposed framework with baselines and state-of-the-art approaches, including individual learning (Mirza & Osindero, 2014), sequential fine-tuning (Wang et al., 2018), multi-task learning (Standley et al., 2020b), FID-transfer learning (Wang et al., 2018), EWC-GAN (Seff et al., 2017), Lifelong-GAN (Zhai et al., 2019), and CAM-GAN (Varshney et al., 2021). The results show the efficacy of our approach in terms of generative performance and the ability to learn new modes while preserving knowledge of the existing modes.

### 4.1 MODE AFFINITY SCORE CONSISTENCY

In the first experiment, the ten generative tasks are defined based on the MNIST dataset, where each task corresponds to generating a specific digit (i.e., $0, 1, \ldots, 9$). For instance, task 0 aimed at generating images representing digit 0, while task 1 aimed at generating images depicting digit 1. The cGAN model was trained to generate images from the nine source tasks while considering one task as the target task. The cGAN model has nine modes corresponding to nine source tasks. The well-trained generator of this cGAN model served as the representation network for the source data. To evaluate the consistency of the closest modes for each target, we conduct 10 trial runs, in which the source cGAN model is initialized randomly. The mean and standard deviation of the mode-affinity scores between each pair of source-target modes are shown in Figure 2 (a) and Figure 6, respectively. In the mean table from Figure 2 (a), the columns denote the distance from each source mode to the given target. For instance, the first column indicates that digits 6 and 9 are closely related to the target digit 0. Similarly, the second column shows that digits 4 and 7 are the closest tasks to the incoming

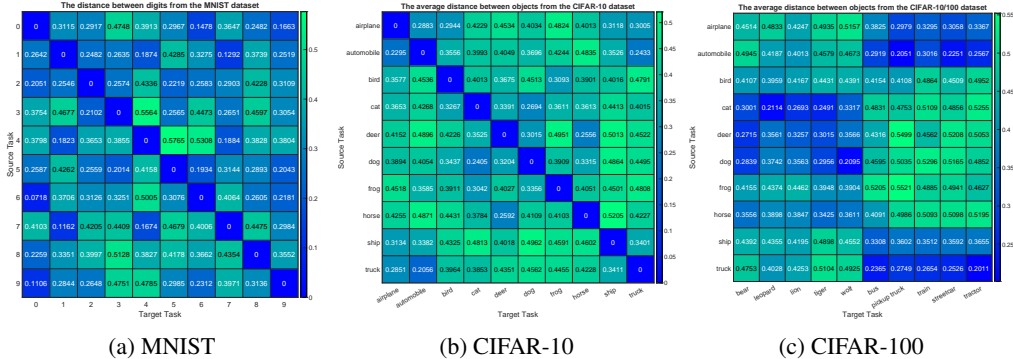

Figure 2: The mean values of the mode-affinity scores computed across 10 trial runs using the cGAN model for data classes from (a) MNIST, (b) CIFAR-10, (c) CIFAR-100 datasets.

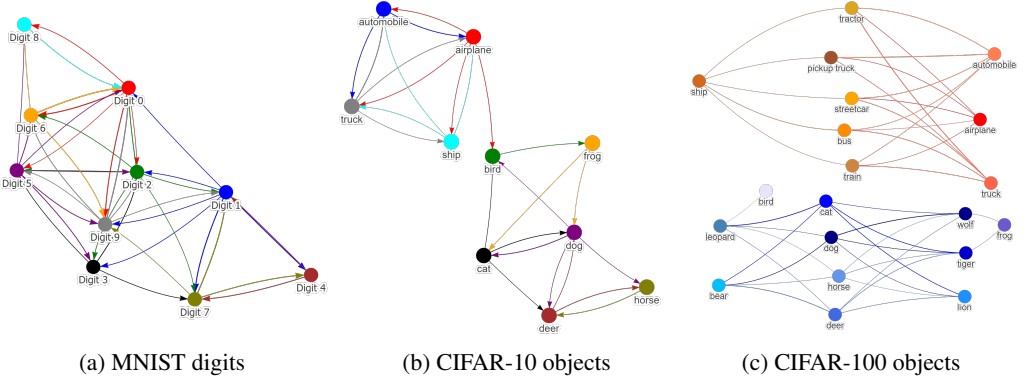

Figure 3: The atlas plot from the computed distances using the cGAN model for data classes from (a) MNIST, (b) CIFAR-10, (c) CIFAR-100 datasets.

digit 1. The standard deviation table from Figure 6 indicates that the calculated distance is stable, as there are no overlapping fluctuations and the orders of similarity between tasks are preserved across 10 runs. In other words, this suggests that the tendency of the closest modes for each target remains consistent regardless of the initialization of cGAN. Thus, the computed mode-affinity score demonstrates consistent results. Additionally, we provide the atlas plot in Figure 3(a) which gives an overview of the relationship between the digits based on the computed distances. The plot reveals that digits $1, 4, 7$ exhibit a notable similarity, while digits $0, 6, 8$ are closely related. This plot provides a useful visualization to showcase the pattern and similarity among different digits.

Analogously, we define ten generative tasks for the CIFAR-10 dataset, each corresponding to a specific object, such as airplane, automobile, bird, cat, deer, dog, frog, horse, ship, and truck. Following the previous experiment, one task is designated as the target task, while the others are the source tasks used to train the cGAN model for image generation. The generator of the well-trained cGAN model serves as the representation network for the source tasks. We present the mean and standard deviation of computed mode-affinity scores between each pair of source-target modes over 10 trial runs in Figure 2(b) and Figure 7, respectively. The mean table in Figure 2(b) shows the average distance of each source mode from the target (e.g., trucks are similar to automobiles, and cats are closely related to dogs). The standard deviation table in Figure 7 demonstrates the stability of the results across different initialization of cGAN. The consistent findings suggest that the computed distances from the CIFAR-10 dataset are reliable. Furthermore, in Figure 3(b), we include the atlas plot which presents an overview of the relationship between the objects based on the computed mode-affinity scores. The plot reveals that automobile, truck, ship, and airplane have a strong connection, while the other classes (i.e., bird, cat, dog, deer, horse, frog) also exhibit a significant resemblance.

Next, the CIFAR-100 dataset is utilized to define ten target tasks, each corresponding to a specific image class, such as bear, leopard, lion, tiger, wolf, bus, pickup truck, train, streetcar, and tractor.

Table 1: Knowledge transfer performance of mode-affinity score against other baselines and FID transfer learning approaches for 10-shot, 20-shot, and 100-shot.

| Approach | Target | 10-shot | 20-shot | 100-shot |
|---|---|---|---|---|
| Individual Learning (Mirza & Osindero, 2014) | Bus | 94.82 | 89.01 | 78.47 |
| Sequential Fine-tuning (Zhai et al., 2019) | Bus | 88.03 | 79.51 | 67.33 |
| Multi-task Learning (Standley et al., 2020b) | Bus | 80.06 | 76.33 | 61.59 |
| FID-Transfer Learning (Wang et al., 2018) | Bus | 61.34 | 54.18 | 46.37 |
| **MA-Transfer Learning (ours)** | **Bus** | **57.16** | **50.06** | **41.81** |

For this experiment, the cGAN model is trained on the entire CIFAR-10 dataset to generate images from ten classes or modes. Figure 2(c) and Figure 8 respectively display the mean and standard deviation of the computed mode-affinity scores between the source-target modes. The mean table in Figure 2(c) indicates the average distance from each CIFAR-10 source mode to the CIFAR-100 target mode. Notably, the target tasks of generating bear, leopard, lion, tiger, and wolf images are closely related to the group of cat, deer, and dog. Specifically, cat images are closely related to leopard, lion, and tiger images. Moreover, the target modes of generating bus, pickup truck, streetcar, and tractor images are highly related to the group of automobile, truck, airplane, and ship. The standard deviation table in Figure 8 also indicates that the computed distances are consistent across different trial runs, demonstrating the stability of the computed mode-affinity scores. Thus, the distances computed between the CIFAR-100 modes and the CIFAR-10 modes are reliable and consistent. In addition, Figure 3(c) includes an atlas plot that provides a visual representation of the relationships between objects based on the computed distances. The plot reveals a strong connection between the vehicles, such as tractors, trucks, and trains, as well as a notable closeness between the animal classes, such as lions, tigers, cats, and dogs. This plot serves as a useful tool for visualizing the similarity and relationship between different objects in the CIFAR-10 and CIFAR-100 datasets and can help to identify relevant data classes for the target class.

Additionally, we conducted a knowledge transfer experiment to assess the effectiveness of the proposed mode affinity score in transfer learning scenarios utilizing the MNIST, CIFAR-10, and CIFAR-100 datasets. In these experiments, we designated one data class as the target, while the remaining nine classes served as source tasks. Our method first computes the dMAS distance from the target to each source task. After identifying the closest task, we fine-tune the cGAN model using the target data samples with the label from the closest task. This approach helps the cGAN model to update the specific part of the model to efficiently and quickly learn the target task. The transfer learning framework is illustrated in Figure 5 and the pseudo-code is provided in Algorithm 2. The image generation performance in terms of FID scores are presented in Tables 1 and 3. Notably, our utilization of dMAS for knowledge transfer consistently outperforms the baseline methods, including Individual Learning, Sequential Fine-tuning, and Multi-task Learning. Remarkably, our approach achieves superior results while utilizing only 10% of the target data samples. When comparing our method with the FID transfer learning approach (Wang et al., 2018), we observe similar performance in most scenarios. However, in the CIFAR-100 experiment, where the target class is bus images, we intentionally trained the source model using only a limited number of truck samples, leading to an inadequately trained source model. Consequently, FID still regards the class of truck images as the closest to the target, resulting in less efficient knowledge transfer. In contrast, our method takes into account the state of the models and selects the task closest to automobile images, resulting in more effective knowledge transfer. Table 1 clearly demonstrates that our approach outperforms FID transfer learning in 10-shot, 20-shot, and 100-shot scenarios.

## 4.2 Continual Learning Performance

We apply the computed mode-affinity scores between generative tasks in the MNIST, CIFAR-10, and CIFAR-100 datasets to the mode-aware continual learning framework. In each dataset, we define two target tasks for continual learning scenarios and consider the remaining eight classes as source tasks. Particularly, (digit 0, digit 1), (truck, cat), and (lion, bus) are the targets for the MNIST, CIFAR-10, and CIFAR-100 experiments, respectively. The cGAN model is trained to sequentially update these target tasks. Here, we select the top-2 closest modes to each target and leverage their

Table 2: Comparison of the mode-aware continual learning framework for cGAN against other baseline and state-of-the-art approaches for MNIST, CIFAR-10, and CIFAR-100, in terms of FID.

| | | MNIST | | |
|---|---|---|---|---|
| **Approach** | **Target** | $\mathcal{P}_{target}$ | $\mathcal{P}_{closest}$ | $\mathcal{P}_{average}$ |
| Individual Learning (Mirza & Osindero, 2014) | Digit 0 | 19.62 | - | - |
| Sequential Fine-tuning (Zhai et al., 2019) | Digit 0 | 16.72 | 26.53 | 26.24 |
| Multi-task Learning (Standley et al., 2020b) | Digit 0 | 11.45 | **5.83** | 6.92 |
| EWC-GAN (Seff et al., 2017) | Digit 0 | 8.96 | 7.51 | 7.88 |
| Lifelong-GAN (Zhai et al., 2019) | Digit 0 | 8.65 | 6.89 | 7.37 |
| CAM-GAN (Varshney et al., 2021) | Digit 0 | 7.02 | 6.43 | 6.41 |
| **MA-Continual Learning (ours)** | **Digit 0** | **6.32** | 5.93 | **5.72** |
| Individual Learning (Mirza & Osindero, 2014) | Digit 1 | 20.83 | - | - |
| Sequential Fine-tuning (Zhai et al., 2019) | Digit 1 | 18.24 | 26.73 | 27.07 |
| Multi-task Learning (Standley et al., 2020b) | Digit 1 | 11.73 | 6.51 | 6.11 |
| EWC-GAN (Seff et al., 2017) | Digit 1 | 9.62 | 8.65 | 8.23 |
| Lifelong-GAN (Zhai et al., 2019) | Digit 1 | 8.74 | 7.31 | 7.29 |
| CAM-GAN (Varshney et al., 2021) | Digit 1 | 7.42 | 6.58 | 6.43 |
| **MA-Continual Learning (ours)** | **Digit 1** | **6.45** | **6.14** | **5.92** |
| | | CIFAR-10 | | |
| **Approach** | **Target** | $\mathcal{P}_{target}$ | $\mathcal{P}_{closest}$ | $\mathcal{P}_{average}$ |
| Individual Learning (Mirza & Osindero, 2014) | Truck | 72.18 | - | - |
| Sequential Fine-tuning (Zhai et al., 2019) | Truck | 61.52 | 65.18 | 64.62 |
| Multi-task Learning (Standley et al., 2020b) | Truck | 55.32 | **33.65** | 35.52 |
| EWC-GAN (Seff et al., 2017) | Truck | 44.61 | 35.54 | 35.21 |
| Lifelong-GAN (Zhai et al., 2019) | Truck | 41.84 | 35.12 | 34.67 |
| CAM-GAN (Varshney et al., 2021) | Truck | 37.41 | 34.67 | 34.24 |
| **MA-Continual Learning (ours)** | **Truck** | **35.57** | 34.68 | **33.89** |
| Individual Learning (Mirza & Osindero, 2014) | Cat | 65.18 | - | - |
| Sequential Fine-tuning (Zhai et al., 2019) | Cat | 61.36 | 67.82 | 65.23 |
| Multi-task Learning (Standley et al., 2020b) | Cat | 54.47 | **34.55** | 36.74 |
| EWC-GAN (Seff et al., 2017) | Cat | 45.17 | 36.53 | 35.62 |
| Lifelong-GAN (Zhai et al., 2019) | Cat | 42.58 | 35.76 | 34.89 |
| CAM-GAN (Varshney et al., 2021) | Cat | 37.29 | 35.28 | 34.62 |
| **MA-Continual Learning (ours)** | **Cat** | **35.29** | 34.76 | **34.01** |
| | | CIFAR-100 | | |
| **Approach** | **Target** | $\mathcal{P}_{target}$ | $\mathcal{P}_{closest}$ | $\mathcal{P}_{average}$ |
| Individual Learning (Mirza & Osindero, 2014) | Lion | 72.58 | - | - |
| Sequential Fine-tuning (Zhai et al., 2019) | Lion | 63.78 | 66.56 | 65.82 |
| Multi-task Learning (Standley et al., 2020b) | Lion | 56.32 | **36.38** | 37.47 |
| EWC-GAN (Seff et al., 2017) | Lion | 46.53 | 38.79 | 36.72 |
| Lifelong-GAN (Zhai et al., 2019) | Lion | 43.57 | 38.35 | 36.53 |
| CAM-GAN (Varshney et al., 2021) | Lion | 40.24 | 37.64 | 36.86 |
| **MA-Continual Learning (ours)** | **Lion** | **38.73** | 36.53 | **35.88** |
| Individual Learning (Mirza & Osindero, 2014) | Bus | 78.47 | - | - |
| Sequential Fine-tuning (Zhai et al., 2019) | Bus | 67.51 | 70.77 | 69.26 |
| Multi-task Learning (Standley et al., 2020b) | Bus | 61.86 | **37.21** | 38.21 |
| EWC-GAN (Seff et al., 2017) | Bus | 49.86 | 39.84 | 37.91 |
| Lifelong-GAN (Zhai et al., 2019) | Bus | 43.73 | 39.75 | 37.66 |
| CAM-GAN (Varshney et al., 2021) | Bus | 42.81 | 38.82 | 37.21 |
| **MA-Continual Learning (ours)** | **Bus** | **41.68** | 38.63 | **36.87** |

knowledge for quick adaptation in learning the target task while preventing catastrophic forgetting. First, we construct a label embedding for the target data samples based on the label embeddings of the top-2 closest modes and the computed distances, as shown in Equation (3). Next, we fine-tune the source cGAN model with the newly-labeled target samples, while also implementing memory replay to avoid catastrophic forgetting of the existing modes. After adding the first target to cGAN,

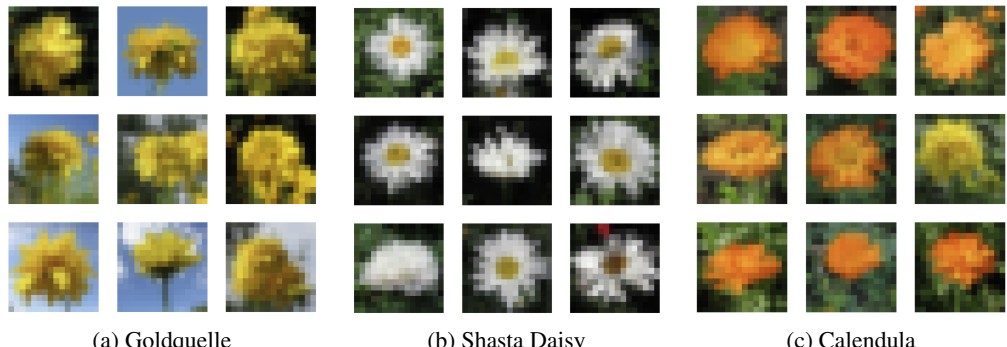

| (a) Goldquelle | (b) Shasta Daisy | (c) Calendula |

Figure 4: The generated image samples from the continual learning cGAN in Oxford Flower dataset, with top-2 relevant modes: (a) Goldquelle, (b) Shasta Daisy, and the target mode: (c) Calendula.

we continue the continual learning process for the second target in each experiment. We compare our framework with sequential fine-tuning (Zhai et al., 2019), multi-task learning (Standley et al., 2020b), EWC-GAN (Seff et al., 2017), lifelong-GAN (Zhai et al., 2019), and CAM-GAN (Varshney et al., 2021) for the few-shot generative task with 100 target data samples. We report the FID scores of the images from the target mode, top-2 closest modes, and the average of all modes in Table 2. By selectively choosing and utilizing the relevant knowledge from learned modes, our approach significantly outperforms the conventional training methods (i.e., sequential fine-tuning, and multi-task learning) for both the first (i.e., digit 0, truck, lion) and the second generative tasks (i.e., digit 1, cat, and bus). The results further demonstrate that our proposed mode-aware continual learning approach significantly outperforms EWC-GAN (Seff et al., 2017) in the second target task in all datasets. Moreover, our model also achieves highly competitive results in comparison to lifelong-GAN (Zhai et al., 2019) and CAM-GAN (Varshney et al., 2021), showcasing its outstanding performance on the first and second target tasks. Although we observed a slight degradation in the performance of the top-2 closest modes due to the trade-off discussed in Theorem 1, our lifelong learning model demonstrates better overall performance when considering all the learned modes.

Moreover, we implement the proposed continual learning framework on the Oxford Flower dataset, specifically focusing on a subset containing ten distinct flower categories. Here, we define ten generative tasks, each corresponding to one of these flower categories. Our objective is to generate images of calendula flowers, thus designating it as the target task. The remaining flower categories, meanwhile, serve as the source tasks for training cGAN. To determine which source tasks share the greatest resemblance to the target task, we employ dMAS to compute the proximity between the target task and each of the source tasks. This analysis has unveiled the two closest tasks to calendula, namely goldquelle and shasta daisy. Hence, we leverage the knowledge from these related tasks to formulate the weighted target label. This label is subsequently employed with calendula samples to fine-tune the cGAN. Figure 4 represents the generated images of the top-2 closest tasks (i.e., goldquelle and shasta daisy), alongside the target task (i.e., calendula), following the fine-tuning process. The results indicate that cGAN effectively leverages the inherent similarity between goldquelle and shasta daisy to enhance its ability to generate calendula flowers. However, it's worth noting that in some instances, the model may generate goldquelle-like images instead of calendula flowers. This occurrence can be attributed to the remarkably close resemblance between these two types of flowers.

## 5  CONCLUSION

We present a new measure of similarity between generative tasks for conditional generative adversarial networks. Our distance, called the discriminator-based mode affinity score, is based on the expectation of the Hessian matrices derived from the discriminator's loss function. This measure provides insight into the difficulty of extracting valuable knowledge from existing modes to learn new tasks. We apply this metric within the framework of continual learning, capitalizing on the knowledge acquired from relevant learned modes to expedite adaptation to new target modes. Through a series of experiments, we empirically validate the efficacy of our approach, highlighting its advantages over traditional fine-tuning methods and other state-of-the-art continual learning techniques.

REPRODUCIBILITY STATEMENT

The detailed task definitions within the datasets are provided in Appendix A, offering a thorough and comprehensive understanding of the experimental framework. Additionally, our selection of continual learning architectures and their corresponding training parameters can be found in both Section 4 and Appendix A. For the transfer learning experiment, we provide a step-by-step procedure (i.e., Figure 5), a pseudo-code (i.e., Algorithm 2), and a comprehensive outline of the experimental setup in Appendix C.1.

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

## A  EXPERIMENTAL SETUP

In this work, we construct $40$ generative tasks based on popular datasets such as MNIST (LeCun et al., 2010), CIFAR-10 (Krizhevsky et al., 2009), CIFAR-100 (Krizhevsky et al., 2009), and Oxford Flower (Nilsback & Zisserman, 2008). For MNIST, we define $10$ distinct generative tasks, each focused on generating a specific digit (i.e., $0, 1, \ldots, 9$). Task 0, for example, is designed to generate the digit $0$, while task 1 generates the digit $1$, and so on. For the CIFAR-10 dataset, we also construct $10$ generative tasks, with each task aimed at generating a specific object category such as airplane, automobile, bird, cat, deer, dog, frog, horse, ship, and truck. Similarly, for the CIFAR-100 dataset, we create $10$ target tasks, each corresponding to a specific image class, including bear, leopard, lion, tiger, wolf, bus, pickup truck, train, streetcar, and tractor. In Oxford Flower dataset, we consider $10$ classes of flowers, including phlox, rose, calendula, iris, shasta daisy, bellflower, viola, goldquelle, peony, aquilegia. Each flower category consists of $80$ image samples. The sample was originally $128 \times 128$, but resized to $16 \times 16$ to reduce the computational complexity.

To represent the generative tasks, we utilize the conditional Wasserstein GAN with Gradient Penalty (cWGAN-GP) model (Gulrajani et al., 2017; Yonekura et al., 2021). In each experiment, we select a specific task as the target task, while considering the other tasks as source tasks. To represent these source tasks, we train the cWGAN-GP model on their respective datasets. This enables us to generate high-quality samples that are representative of the source tasks. Once trained, we can use the cWGAN-GP model as the representation network for the generative tasks. This model is then applied to our proposed mode-aware continual learning framework. We compare our method against several approaches, including individual learning (Mirza & Osindero, 2014), sequential fine-tuning (Wang et al., 2018), multi-task learning (Standley et al., 2020b), EWC-GAN (Seff et al., 2017), Lifelong-GAN (Zhai et al., 2019), and CAM-GAN (Varshney et al., 2021). Individual learning (Mirza & Osindero, 2014) involves training the cGAN model on a specific task in isolation. In sequential fine-tuning (Wang et al., 2018), the cGAN model is trained sequentially on source and target tasks. Multi-task learning (Standley et al., 2020b), on the other hand, involves training a cGAN model on a joint dataset created from both the source and target tasks. Our method is designed to improve on these approaches by enabling the continual learning of generative tasks while mitigating catastrophic forgetting.

## B  THEORETICAL ANALYSIS

We first recall the definition of the GAN's discriminator loss as follows:

**Definition 2** (Discriminator Loss). *Let $x = \{x_1, \ldots, x_m\}$ be the real data samples, $z$ denote the random vector, and $\theta_{\mathcal{D}}$ be the discriminator's parameters. $\mathcal{D}$ is trained to maximize the probability*

*of assigning the correct label to both training real samples and generated samples $\mathcal{G}(z)$ from the generator $\mathcal{G}$. The objective of the discriminator is to maximize the following function:*

$$\nabla_{\theta_{\mathcal{D}}} \sum_{i=1}^{m} \left[ \log \mathcal{D}\left(x^{(i)}\right) + \log\left(1 - \mathcal{D}\left(\mathcal{G}\left(z^{(i)}\right)\right)\right) \right] \tag{5}$$

We recall the definition of Fisher Information matrix (FIM) (Le et al., 2022a) as follows:

**Definition 3** (Fisher Information). *Given dataset $X$, let $N$ denote a neural network with weights $\theta$, and the negative log-likelihood loss function $L(\theta) := L(\theta, X)$. FIM is described as follows:*

$$F(\theta) = \mathbb{E}\left[\nabla_\theta L(\theta) \nabla_\theta L(\theta)^T\right] = -\mathbb{E}\left[H\left(L(\theta)\right)\right] \tag{6}$$

Next, we present the proof of Theorem 1.

**Theorem 1.** *Let $X_a$ be the source data, characterized by the density function $p_a$. Let $X_b$ be the data for the target mode with data density function $p_b$, $p_b \neq p_a$. Let $\theta$ denote the model's parameters. Consider the loss functions $L_a(\theta) = \mathbb{E}[l(X_a; \theta)]$ and $L_b(\theta) = \mathbb{E}[l(X_b; \theta)]$. Assume that both $L_a(\theta)$ and $L_b(\theta)$ are strictly convex and possess distinct global minima. Let $X_n$ denote the mixture data of $X_a$ and $X_b$ described by $p_n = \alpha p_a + (1 - \alpha)p_b$, where $\alpha \in (0, 1)$. The corresponding loss function is given by $L_n(\theta) = \mathbb{E}[l(X_n; \theta)]$. Under these assumptions, it follows that $\theta^* = \arg\min_\theta L_n(\theta)$ satisfies:*

$$L_a(\theta^*) > \min_\theta L_a(\theta) \tag{7}$$

***Proof of Theorem 1.*** Assume toward contradiction that $L_a(\theta^*) > \min_\theta L_a(\theta)$ does not hold. Because $L_a(\theta^*) \geq \min_\theta L_a(\theta)$ always holds, we must have that:

$$L_a(\theta^*) = \min_\theta L_a(\theta). \tag{8}$$

By the linearity of expectation, we have that:

$$L_n(\theta) = \alpha L_a(\theta) + (1 - \alpha)L_b(\theta)$$

Hence, we have

$$\min_\theta L_n(\theta) = L_n(\theta^*)$$
$$= \alpha L_a(\theta^*) + (1 - \alpha)L_b(\theta^*)$$
$$= \alpha \min_\theta L_a(\theta) + (1 - \alpha)L_b(\theta^*)$$
$$> \alpha \min_\theta L_a(\theta) + (1 - \alpha)L_a(\theta^*)$$
$$= \alpha \min_\theta L_a(\theta) + (1 - \alpha)\min_\theta L_a(\theta)$$
$$= \min_\theta L_a(\theta)$$

where in the third equality we use the facts that both $L_a$ and $L_b$ are strongly convex and have different global minimum. Because $L_a$ and $L_b$ have the same optimal value (assumed to be 0) and that $\theta^*$ is not the optimal point for $L_b$, we must have $L_b(\theta^*) > L_b(\theta_b^*) = L_a(\theta^*)$ where $\theta_b^* = \arg\min_\theta L_b(\theta)$.

Therefore, we have proved that $\min_\theta L_n(\theta) = L_n(\theta^*) > \min_\theta L_a(\theta)$, contradicting to Eq. equation 8. $\square$

## C  ABLATION STUDIES

### C.1  MODE-AWARE TRANSFER LEARNING

We apply the proposed mode-affinity score to transfer learning in an image generation scenario. The proposed similarity measure enables the identification of the closest modes or data classes to support the learning of the target mode. Here, we introduce a *mode-aware transfer learning* framework that quickly adapts a pre-trained cGAN model to learn the target mode. The overview of the transfer

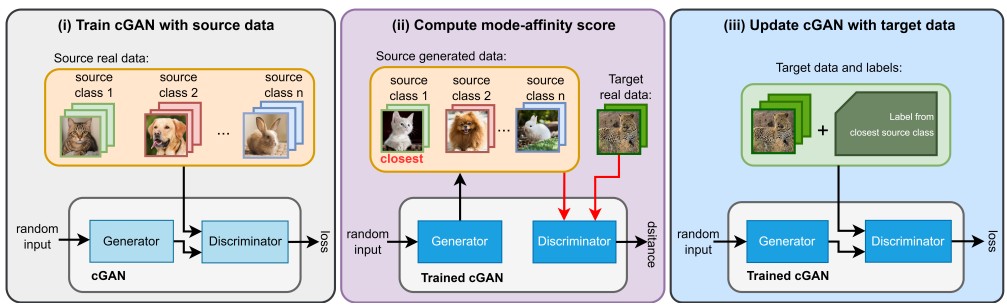

Figure 5: The overview of mode-aware transfer learning framework for the conditional Generative Adversarial Network: (i) Representing source data classes using cGAN, (ii) Computing the mode-affinity from each source mode to the target, (iii) Fine-tuning the generative model using the target data and the label of the closest mode for transfer learning.

---

**Algorithm 2:** Mode-Aware Transfer Learning for Conditional Generative Adversarial Networks

---

**Data:** Source data: $(X_{train}, y_{train})$, Target data: $X_{target}$
**Input:** The generator $\mathcal{G}$ and discriminator $\mathcal{D}$ of cGAN
**Output:** Target generator $\mathcal{G}_{\bar{\theta}}$

1 **Function** dMAS $(X_a, y_a, X_b, \mathcal{G}, \mathcal{D})$ **:**

2      Generate data $\tilde{X}_a$ of class label $y_a$ using the generator $\mathcal{G}$

3      Compute $H_a$ from the loss of discriminator $\mathcal{D}$ using $\{X_a, \tilde{X}_a\}$

4      Compute $H_b$ from the loss of discriminator $\mathcal{D}$ using $\{X_b, \tilde{X}_a\}$

5      **return** $s[a,b] = \dfrac{1}{\sqrt{2}} \left\| H_a^{1/2} - H_b^{1/2} \right\|_F$

6 **Function** Main **:**

7      Train $(\mathcal{G}_\theta, \mathcal{D}_\theta)$ with $X_{train}, y_{train}$          ▷ Pre-train cGAN model

8      Construct S source modes, each from a data class in $y_{train}$

9      **for** $i = 1, 2, \ldots, S$ **do**

10          $s_i = \text{dMAS}(X_{train_i}, y_{train_i}, X_{target}, \mathcal{G}_\theta, \mathcal{D}_\theta)$      ▷ Find the closest modes

11      **return** closest mode(s): $i^* = \underset{i}{\text{argmin}}\, s_i$

                                   ▷ Fine-tune with the target task

12      **while** $\theta$ *not converged* **do**

13          Update $\mathcal{G}_\theta, \mathcal{D}_\theta$ using real data $X_{target}$ and closest source label $y_{train_{i^*}}$

14      **return** $\mathcal{G}_{\bar{\theta}}$

---

learning framework is illustrated in Figure 5. Particularly, we select the closest source mode from the pool of multiple learned modes based on the computed dMAS.

To leverage the knowledge of the closest mode for training the target mode, we assign the target data samples with labels of the closest mode. Subsequently, we use these modified target data samples to fine-tune the generator and discriminator of the pre-trained cGAN model. Figure 5(3) illustrates the transfer learning method, where the data class 1 (i.e., cat images) is the most similar to the target data (i.e., leopard image) based on the computed dMAS. Hence, we assign the label of class 1 to the leopard images. The pre-trained GAN model uses this modified target data to quickly adapt the cat image generation to the leopard image generation. The mode-aware algorithm for transfer learning in cGAN is described in Algorithm 2. By assigning the closest mode's label to the target data samples, our method can effectively fine-tune the relevant parts of cGAN for learning the target mode. This approach helps improve the training process and reduces the number of required target training data.

Next, we conduct experiments employing mode affinity scores within the context of transfer learning scenarios. These experiments were designed to assess the effectiveness of our proposed mode-affinity measure in the transfer learning framework. In this scenario, each generative task corresponds to a single data class within the MNIST (LeCun et al., 2010), CIFAR-10 (Krizhevsky et al., 2009), and

Table 3: Comparison of the mode-aware transfer learning framework for cGAN against other baselines and FID-transfer learning approach in terms of FID.

| Approach | Target | MNIST 10-shot | 20-shot | 100-shot |
|---|---|---|---|---|
| Individual Learning (Mirza & Osindero, 2014) | Digit 0 | 34.25 | 27.17 | 19.62 |
| Sequential Fine-tuning (Zhai et al., 2019) | Digit 0 | 29.68 | 24.22 | 16.14 |
| Multi-task Learning (Standley et al., 2020b) | Digit 0 | 26.51 | 20.74 | 10.95 |
| FID-Transfer Learning (Wang et al., 2018) | Digit 0 | **12.64** | **7.51** | **5.53** |
| **MA-Transfer Learning (ours)** | **Digit 0** | **12.64** | **7.51** | **5.53** |
| Individual Learning (Mirza & Osindero, 2014) | Digit 1 | 35.07 | 29.62 | 20.83 |
| Sequential Fine-tuning (Zhai et al., 2019) | Digit 1 | 28.35 | 24.79 | 15.85 |
| Multi-task Learning (Standley et al., 2020b) | Digit 1 | 26.98 | 21.56 | 10.68 |
| FID-Transfer Learning (Wang et al., 2018) | Digit 1 | **11.35** | **7.12** | **5.28** |
| **MA-Transfer Learning (ours)** | **Digit 1** | **11.35** | **7.12** | **5.28** |
| Approach | Target | CIFAR-10 10-shot | 20-shot | 100-shot |
| Individual Learning (Mirza & Osindero, 2014) | Truck | 89.35 | 81.74 | 72.18 |
| Sequential Fine-tuning (Zhai et al., 2019) | Truck | 76.93 | 70.39 | 61.41 |
| Multi-task Learning (Standley et al., 2020b) | Truck | 72.06 | 65.38 | 55.29 |
| FID-Transfer Learning (Wang et al., 2018) | Truck | **51.05** | **44.93** | **36.74** |
| **MA-Transfer Learning (ours)** | **Truck** | **51.05** | **44.93** | **36.74** |
| Individual Learning (Mirza & Osindero, 2014) | Cat | 80.25 | 74.46 | 65.18 |
| Sequential Fine-tuning (Zhai et al., 2019) | Cat | 73.51 | 68.23 | 59.08 |
| Multi-task Learning (Standley et al., 2020b) | Cat | 68.73 | 61.32 | 50.65 |
| FID-Transfer Learning (Wang et al., 2018) | Cat | **47.39** | **40.75** | **32.46** |
| **MA-Transfer Learning (ours)** | **Cat** | **47.39** | **40.75** | **32.46** |
| Approach | Target | CIFAR-100 10-shot | 20-shot | 100-shot |
| Individual Learning (Mirza & Osindero, 2014) | Lion | 87.91 | 80.21 | 72.58 |
| Sequential Fine-tuning (Zhai et al., 2019) | Lion | 77.56 | 70.76 | 61.33 |
| Multi-task Learning (Standley et al., 2020b) | Lion | 71.25 | 67.84 | 56.12 |
| FID-Transfer Learning (Wang et al., 2018) | Lion | **51.08** | **46.97** | **37.51** |
| **MA-Transfer Learning (ours)** | **Lion** | **51.08** | **46.97** | **37.51** |
| Individual Learning (Mirza & Osindero, 2014) | Bus | 94.82 | 89.01 | 78.47 |
| Sequential Fine-tuning (Zhai et al., 2019) | Bus | 88.03 | 79.51 | 67.33 |
| Multi-task Learning (Standley et al., 2020b) | Bus | 80.06 | 76.33 | 61.59 |
| FID-Transfer Learning (Wang et al., 2018) | Bus | 61.34 | 54.18 | 46.37 |
| **MA-Transfer Learning (ours)** | **Bus** | **57.16** | **50.06** | **41.81** |

CIFAR-100 (Krizhevsky et al., 2009) datasets. Here, in our transfer learning framework, we leverage the computed mode-affinity scores between generative tasks. Specifically, we utilize this distance metric to identify the mode closest to the target mode and then fine-tune the conditional Generative Adversarial Network (cGAN) accordingly. To achieve this, we assign the target data samples with the labels of the closest mode and use these newly-labeled samples to train the cGAN model. By doing so, the generative model can benefit from the knowledge acquired from the closest mode, enabling quick adaptation in learning the target mode. In this study, we compare our proposed transfer learning framework with several baselines and state-of-the-art approaches, including individual learning (Mirza & Osindero, 2014), sequential fine-tuning (Wang et al., 2018), multi-task learning (Standley et al., 2020b), and FID-transfer learning (Wang et al., 2018). Additionally, we present a performance comparison of our mode-aware transfer learning approach with these methods for 10-shot, 20-shot, and 100-shot scenarios in the MNIST, CIFAR-10, and CIFAR-100 datasets (i.e., the target dataset contains only 10, 20, or 100 data samples).

Across all three datasets, our results demonstrate the effectiveness of our approach in terms of generative performance and its ability to efficiently learn new tasks. Our proposed framework significantly outperforms individual learning and sequential fine-tuning while demonstrating strong

Table 4: Comparison of the mode-aware continual learning performance between different choices of the number of closest modes

| Approach | Dataset | Target | Performance |
|---|---|---|---|
| MA-Continual Learning with top-2 closest modes | MNIST | Digit 0 | 6.32 |
| **MA-Continual Learning with top-3 closest modes** | **MNIST** | **Digit 0** | **6.11** |
| MA-Continual Learning with top-4 closest modes | MNIST | Digit 0 | 6.78 |
| MA-Continual Learning with top-2 closest modes | CIFAR-10 | Truck | 35.57 |
| **MA-Continual Learning with top-3 closest modes** | **CIFAR-10** | **Truck** | **35.52** |
| MA-Continual Learning with top-4 closest modes | CIFAR-10 | Truck | 36.31 |
| MA-Continual Learning with top-2 closest modes | CIFAR-100 | Lion | 38.73 |
| MA-Continual Learning with top-3 closest modes | CIFAR-100 | Lion | 38.54 |
| **MA-Continual Learning with top-4 closest modes** | **CIFAR-100** | **Lion** | **38.31** |

performance even with fewer samples compared to multi-task learning. Moreover, our approach is competitive with FID transfer learning, where the similarity measure between generative tasks is based on FID scores. Notably, our experiments with the CIFAR-100 dataset reveal that FID scores may not align with intuition and often result in poor performance. Notably, for the MNIST dataset, we consider generating digits 0 and 1 as the target modes. As shown in Table 3, our method outperforms individual learning, sequential fine-tuning, and multi-task learning approaches significantly, while achieving similar results compared with the FID transfer learning method. Since the individual learning model lacks training data, it can only produce low-quality samples. On the other hand, the sequential fine-tuning and multi-task learning models use the entire source dataset while training the target mode, which results in better performance than the individual learning method. However, they cannot identify the most relevant source mode and data, thus, making them inefficient compared with our proposed mode-aware transfer learning approach. In other words, the proposed approach can generate high-quality images with fewer target training samples. Notably, the proposed approach can achieve better results using only 20% of data samples. For more complex tasks, such as generating cat and truck images in CIFAR-10 and lion and bus images in CIFAR-100, our approach achieves competitive results to other methods while requiring only 10% training samples. Hence, the mode-aware transfer learning framework using the Discriminator-based Mode Affinity Score can effectively identify relevant source modes and utilize their knowledge for learning the target mode.

## C.2 CHOICE OF CLOSEST MODES

In this experiment, we evaluate the effectiveness of our proposed continual learning framework by varying the number of closest existing modes used for fine-tuning the target mode. Throughout this paper, we opt to utilize the top-2 closest modes, a choice driven by its minimal computational requirements. Opting for a single closest mode (i.e., transfer learning scenarios) would essentially replace that mode with the target mode, negating the concept of continual learning. Here, we explore different scenarios across the MNIST, CIFAR-10, and CIFAR-100 datasets, where we investigate the top-2, top-3, and top-4 closest modes for continual learning. As detailed in Table 4, selecting the three closest modes yields the most favorable target generation performance in the MNIST and CIFAR-10 experiments. Notably, knowledge transfer from the four closest modes results in the weakest performance. This discrepancy can be attributed to the simplicity of these datasets and their highly distinguishable data classes. In such cases, employing more tasks resembles working with dissimilar tasks, leading to negative transfer during target mode training. Conversely, in the CIFAR-100 experiment, opting for the top-4 modes yields the best performance. This outcome stems from the dataset's complexity, where utilizing a larger set of relevant modes confers an advantage during the fine-tuning process.

In summary, the choice of the top-N closest modes is highly dependent on the dataset and available computational resources. Employing more modes necessitates significantly more computational resources and training time for memory replay of existing tasks. It's crucial to note that with an increased number of related modes, the model requires more time and data to converge effectively.

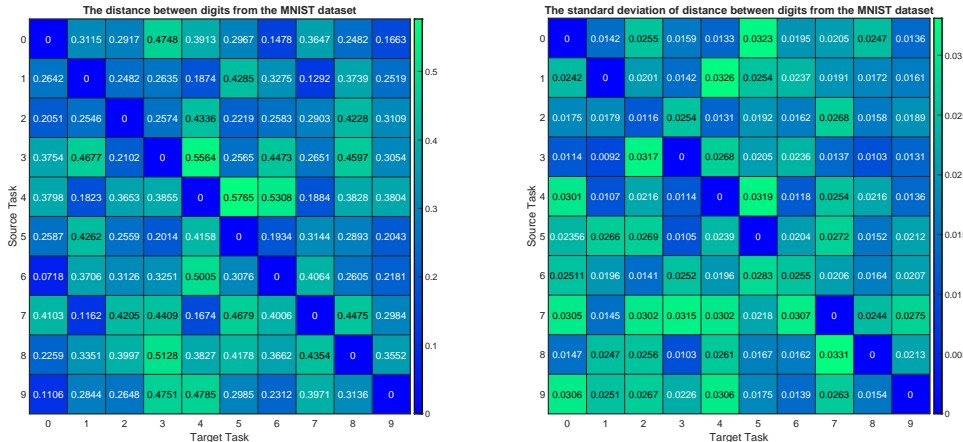

Figure 6: The mean (left) and standard deviation (right) of computed mode-affinity scores between data classes (i.e., digits $0, 1, \ldots, 9$) of the MNIST dataset using cGAN.

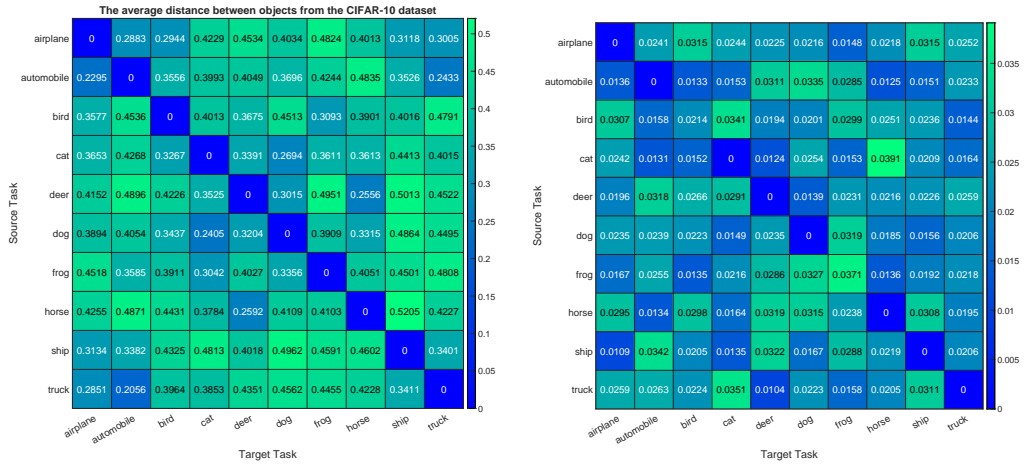

Figure 7: The mean (left) and standard deviation (right) of computed mode-affinity scores between data classes (i.e., airplane, automobile, bird,..., truck) of the CIFAR-10 dataset using cGAN.

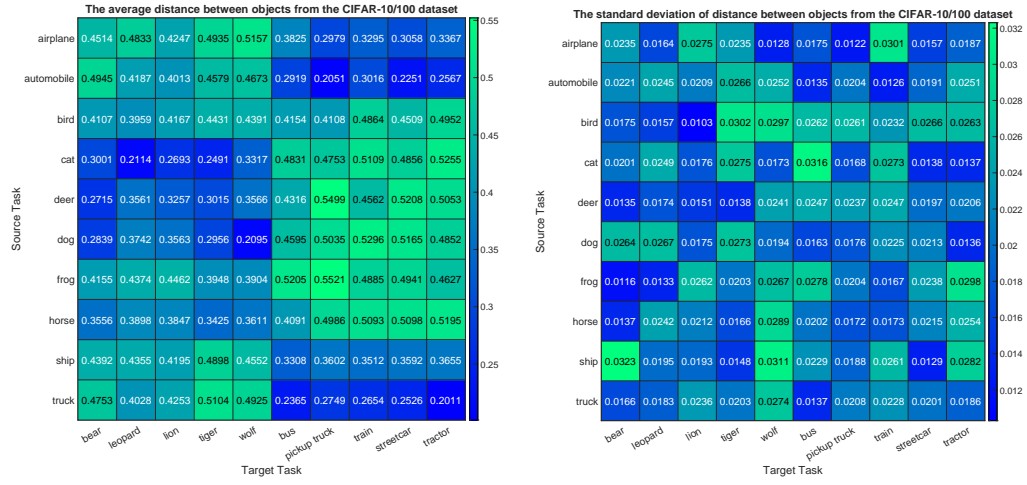

Figure 8: The mean (left) and standard deviation (right) of computed mode-affinity scores between data classes of the CIFAR-10 and CIFAR-100 datasets using cGAN.

## C.3   MODE AFFINITY SCORE CONSISTENCY

To rigorously evaluate and demonstrate the statistical significance of our proposed dMAS in the context of the MNIST, CIFAR-10, and CIFAR-100 datasets, we conducted a comprehensive series of distance consistency experiments. This experimental approach was designed to validate the effectiveness and reliability of the dMAS metric in assessing the affinity between generative tasks within the realm of conditional Generative Adversarial Networks (cGANs). We first initiated each experiment by computing the dMAS distance between different tasks, ensuring a diverse range of task combinations to comprehensively test the metric's robustness. Importantly, we repeated each experiment a total of 10 times, introducing variability through distinct initialization settings for pre-training the cGAN model. This multi-run approach was adopted to account for any potential variability introduced by the initialization process and to capture the metric's performance across a spectrum of scenarios.

We calculated both the mean and standard deviation values of the dMAS distances obtained from each of these 10 experimental runs. This analysis allowed us to quantitatively assess the central tendency and variability of the dMAS measurements. To visually convey the results and facilitate a clear understanding of our findings, we illustrated the mean and standard deviation values of the dMAS distances in Figure 6 for the MNIST dataset, Figure 7 for CIFAR-10, and Figure 8 for CIFAR-100. These figures provide a visual representation of the consistency and stability of the dMAS metric across different tasks and initialization settings. In summary, our experiments and comprehensive analysis conclusively establish the dMAS as a consistently reliable distance metric for assessing the affinity between generative tasks within the framework of cGANs. These findings underscore the metric's robustness and its potential utility in various generative modeling applications.

