# OpenReview forum: "Mode-Aware Continual Learning for Conditional Generative Adversarial Networks"
_ICLR.cc/2024/Conference — Submitted to ICLR 2024_

### Official Review · Reviewer_M2Vt · 2023-10-30

**Soundness:** 2 fair
**Presentation:** 1 poor
**Contribution:** 1 poor
**Rating:** 3
**Confidence:** 5

**Summary:**

This paper studies an interesting topic in continual learning, aiming to train a conditional GAN without forgetting. The main idea of this work is to develop a new discriminator-based mode-affinity measure that can evaluate the similarity between modes. The experiment results on several datasets have demonstrated that the proposed approach achieves promising results.

**Strengths:**

1. The introduction section of this paper is written well.
2. The proposed approach is reasonable.

**Weaknesses:**

1. The main contribution in this paper is very small. This work just proposes a discriminator-based mode-affinity measure, which is a natural choice.
2. In the introduction section, the primary motivation of introducing Discriminator-based Mode Affinity Score is lacking. Why do we need such an approach?
3. The notations X_a in this paper are not clear to me. These notations should be bold because they are matrixes.
4. This paper employs the conditional GAN. However, I do not find the actual loss functions as well as the model in the text.
5. In algorithm 2, the definition of various models, such as G_θ is not defined in the paper. The whole methodology section is hard to follow since it misses some important information.
6. The proposed approach relies on class and task information, which can not be used in a more realistic continual learning setting such as unsupervised learning.
7. Why use the conditional GAN instead of other models such as WGAN?
8. In theorem 1, some notations are not defined or explained. For example, what is "
"trace" in Eq.1. What is $|| ||_F$ in Eq.1?
9. To avoid forgetting, this work employs the generative replay mechanism, which has been done in a wide range of works.
10. The whole algorithm 1 is unclear to me because a lot of definitions are not explained. For example, $S$ in algorithm 1 is not described in the text.
11. The main objective function and the models are not defined and described in the text, which makes it difficult for the readers to understand the main contribution.

Overall, this paper is hard to follow because a lot of notations are not described clearly. The whole methodology section does not clearly describe the actual algorithm and model.

**Questions:**

Please see the weakness section.

---

> ### Author Response · Authors · 2023-11-15
> **Response to Reviewer M2Vt**
>
> Apologies for any confusion. It seems there might be a slight misunderstanding regarding the central concept of our paper.
> * Primarily, our research introduces a groundbreaking task affinity measure tailored for generative tasks within conditional Generative Adversarial Networks (cGANs). Additionally, we present a novel distance-based continual learning framework designed to refine cGAN models.
> * Our motivation behind incorporating this distance-based approach lies in the necessity to leverage pre-existing well-trained models for rapid adaptation to new tasks. To illustrate, consider a model proficient in generating diverse types of cancers. Our objective is to continually enhance its training as new data becomes available, ensuring its adaptability and continual improvement.
> * We employ a conditional Generative Adversarial Network (cGAN) to address the specific challenge of generating images aligned with corresponding labels. It's crucial to note that our focus is not on tackling unsupervised learning problems in this context. The choice of utilizing a conditional Wasserstein GAN (cWGAN) is deliberate; we aim to generate images based on provided labels. For instance, when given the label "lion," our model is intended to generate lion images, distinguishing itself from standard GANs that generate image classes randomly.
> * We leverage a conditional Generative Adversarial Network (cGAN), referencing the pertinent paper for further details. Given constraints on page length, these common practices are omitted from the main paper. In our notation, G_θ represents the generator of the cGAN, parameterized by θ. This paper contributes a comprehensive framework applicable to various types of conditional GANs. It's important to note that our approach is versatile, accommodating any conditional GAN model. For a deeper understanding, additional specifics can be found in the standard conditional GAN paper referenced herein.
> * We employ a standard and straightforward notation in Theorem 1. In this context, "trace" denotes the trace of a matrix, and
> ∥∥F signifies the Frobenius norm.
> * The replay mechanism serves as an additional strategy for mitigating catastrophic forgetting, aimed at enhancing overall performance. We acknowledge that it is not presented as a novel contribution to our work. Its incorporation is motivated by the prevalent usage of such mechanisms in many continual learning approaches, similar to techniques like distillation, highlighting its common adoption in the field.
> * In Algorithm 1, the set S is explicitly defined as the collection of source tasks within the for loop. This notation adheres to the common practice in pseudocode writing.
> * As clearly stated in Theorem 1, X_a denotes the input data (i.e., images). Importantly, it should be noted that X_a is NOT a matrix; rather, it can consist of one or more images employed in the distance computation process.
> * At the end of the Introduction section, we explicitly stated our contribution.

---

### Official Review · Reviewer_cors · 2023-10-31

**Soundness:** 3 good
**Presentation:** 3 good
**Contribution:** 3 good
**Rating:** 5
**Confidence:** 4

**Summary:**

This work tackles the task of continual learning in class-conditional GANs (cGANs). The method consists of two main contributions. In the first part, the authors propose a method to measure the affinity between the classes in a pretrained cGAN and a target class based on the Fisher Information distance. In the second part, the authors use the obtained affinity scores to form the target class embedding as the weighted sum of the most similar source classes. The authors evaluate their proposed method on different datasets in two setups: transfer learning and continual learning.

**Strengths:**

-- The paper is well-written.

-- The proposed method for measuring the affinity between classes is interesting and novel.

-- The experiments show the effectiveness and consistency  of the proposed affinity score in identifying the most similar classes

-- Based on the provided results, the incorporation of the proposed score in transfer and continual learning appears to be effective compared to the baselines.

**Weaknesses:**

-- The idea of class-specific knowledge transfer in conditional GANs has been previously explored in cGANTransfer[1] by learning the class affinities. A discussion of the work and how it compares with the proposed method would improve the completeness of the paper. [2] is another relevant work that could be discussed in the paper.

-- To complete the experiments and to better show the advantage of their proposed affinity score, authors could include some comparison with other affinity metrics such as FID.

-- Although the proposed method has been evaluated on several datasets, it would be better if more complex datasets such as Imagenet were included in the experiments.

-- In the continual learning setup, only two classes are used as targets in each trial. For a more realistic setup, more target classes might be needed in the evaluations

[1] Shahbazi et. al., "Efficient Conditional GAN Transfer with Knowledge Propagation across Classes," CVPR 2021.
[2] Careil et. al., "Few-shot Semantic Image Synthesis with Class Affinity Transfer," CVPR 2023.

**Questions:**

-- Is the target embedding obtained using the class affinity fixed in the proposed method, or is it also fine-tuned with the rest of the generator? what is the reason for such a choice?

-- In section 4.1, the authors mention they initialize the source cGAN randomly. By initialization, do they mean weight initialization or the classes used as the source modes?

-- How does the method compare to the baselines, if there are no semantically similar classes in the source model?

---

> ### Author Response · Authors · 2023-11-19
> **Response to Reviewer cors**
>
> Thank you for your valuable feedback. We appreciate your suggestion to consider the noteworthy papers on knowledge transfer in cGANs. In our revised version, we will certainly incorporate a thorough comparison of these techniques to enrich the depth of our analysis.
>
> We've compared our approach with the FID approach, as shown in Table 2, referred to as Sequential Fine-tuning (Zhai et al., 2019). This method employs FID to discern relevant tasks, followed by fine-tuning based on the identified related task. This comparison provides a valuable perspective on our approach.
>
> We are committed to enhancing the clarity and comprehensiveness of our work based on your feedback. If you have any further suggestions or insights, we welcome the opportunity to address them in our revised version. Thank you once again for your thoughtful input.

---

> > ### Comment · Reviewer_cors · 2023-11-20
> >
> > It is not clear to me how the baseline "Sequential Fine-tuning (Zhai et al., 2019)" is using FID to determine class affinities. In my understanding, the referred baseline approaches lifelong learning with knowledge distillation, without incorporating class similarities. I would appreciate further explanation on this.
> >
> > On a related note, the term "Sequential Fine-tuning" has been cited inconsistently throughout the paper, sometimes by (Wang et al., 2018) and sometimes by (Zhai et al., 2019). In my opinion, the term Sequential Fine-tuning describes the method in (Wang et al., 2018) better than the one in (Zhai et al., 2019).
> >
> > My concerns regarding the complexity of the evaluated datasets and the experimental setup (number of new classes in the continual learning setup) remain unresolved. In addition, I would appreciate the author's feedback on the questions asked in the initial review.

---

> > > ### Author Response · Authors · 2023-11-20
> > > **Reponse to Reviewer cors (cont')**
> > >
> > > Thank you for providing additional feedback. The works by Wang et al., 2018, and Zhai et al., 2019, both utilize FID scores as their foundation. Wang et al. specifically addressed transfer learning, while Zhai et al. tackled the continual learning problem. In the process of sequential fine-tuning, when a new task emerges, our model initially identifies the closest learned tasks by employing the FID metric. Subsequently, it learns the new tasks by leveraging the knowledge acquired from the relevant learned tasks.
> > >
> > > In response to Reviewer cors' inquiries, we would like to address the following points:
> > >
> > > * The embeddings remain fixed during distance calculation to utilize the existing generator and discriminator for identifying the closest learned tasks. After identifying these tasks, we create a weighted label for the target and fine-tune both the generator and discriminator using the target data and label. In this case, the embeddings of the target are not fixed.
> > >
> > > * In each run, the weight of the conditional Generative Adversarial Network (cGAN) is randomly initialized. This ensures that the model's weights differ across trial runs after training with source tasks. We calculate the distance for each run, demonstrating that the obtained distances between tasks exhibit a consistent trend. This experiment underscores the stability of our proposed distance-based approach.
> > >
> > > * The distance-based method takes into account the current state of the model. If the model is poorly trained on a specific task, the distance metric captures and reflects this issue. In contrast, FID scores do not consider the model's state, making it challenging to capture the current performance. Moreover, FID scores are computed based on the Inception v3 model, primarily trained on the standard ImageNet dataset. Consequently, if the target task involves non-standard objects or topics, the FID score may lose relevance. For instance, in the case of a target task like Electron Backscatter Diffraction (EBSD) images of iron, indicating the microstructure orientation, FID scores would not be applicable as they are not designed for such specialized tasks.

---

### Official Review · Reviewer_y3EF · 2023-11-01

**Soundness:** 2 fair
**Presentation:** 2 fair
**Contribution:** 2 fair
**Rating:** 5
**Confidence:** 5

**Summary:**

The authors tackle continual image generation, aiming to identify similar mode for target mode for accelerated learning while preventing catastrophic forgetting. They introduce the Discriminator-based Mode Affinity Score, utilizing the Hessian matrix of the discriminator loss w.r.t. images from each mode. This affinity score aids in comparing the target mode with existing ones, assigning a pseudo label to the target mode. The method leverages target data, labels, and replay data from the source to fine-tune GANs. Theoretically, the authors prove that the performance of existing modes remains unaffected upon integrating a new mode. Empirically, their method surpasses current techniques on CIFAR-10, CIFAR-100, and Oxford-flowers datasets, showcasing its efficacy.

**Strengths:**

1. The exploration of GANs within continual learning for image generation is a compelling research topic.

2. The authors introduce affinity scores derived from the Hessian matrix, which is new.

3. The authors demonstrate that their method outperforms baseline models by conducting experiments on 3 datasets.

**Weaknesses:**

1. The validity of the proposed affinity scores enhancing continual learning’s effectiveness remains unclear. Despite Section 3.2’s assertion that "our measure aligns more closely with human intuition and consistently demonstrates its reliability", the paper lacks empirical or theoretical analysis to substantiate this claim.

2. The employment of a memory replay technique to prevent catastrophic forgetting is not novel, as it is contribution from existing work and thus does not contribute to the originality of this research.

3. Theorem 1 merely establishes that the integration of a new mode will not enhance the performance of existing modes, without providing insight into why the proposed method excels. Thus, Theorem 1 provides no positive roles for enhancing the soundness of this work.

4. The paper lacks a quantitative assessment of performance on the Oxford-flowers dataset, making it difficult to gauge the method's effectiveness in that context.

5. The textual quality and logical coherence of Section 4 are weak. It would be better to reorganize section 4, make it more clear and concise.

**Questions:**

1. two "between"s in "our proposed dMAS quantifies the Fisher Information distance between between the model weights" in Sec 3.2.

---

> ### Author Response · Authors · 2023-11-19
> **Response to Reviewer y3EF**
>
> Thank you for providing constructive feedback. Below are our responses addressing your concerns:
>
> * The proposed task affinity score is based on Fisher Information, a method proven effective in modeling image classification tasks [1] and their proximity. Figure 8 in the Appendix illustrates the task distances in MNIST and CIFAR-10, aligning with both classification tasks' proximity and human intuition. Due to space constraints, we omitted additional ablation studies comparing results, but we commit to incorporating a detailed task distance analysis in the revised manuscript.
>
> * Theorem 1 establishes that introducing a new class of data into a well-trained model leads to a performance decline in existing tasks. Our results in Table 2 consistently demonstrate this trade-off, even with highly similar data. Our objective is to minimize catastrophic forgetting and prevent degradation in learned tasks. The theorem motivates us to learn the new task while preserving the performance of the old tasks. It underscores our focus on not enhancing the learned tasks' performance, as doing so would necessitate sacrificing the new task.
>
> * The replay mechanism is an additional strategy to alleviate catastrophic forgetting, enhancing overall performance. We acknowledge that it is not a novel contribution but is included due to its widespread use in continual learning approaches. This parallels techniques like distillation, emphasizing its common adoption in the field.
>
> * We apologize for any confusion in Section 4 and commit to revising it for clearer motivation. Our aim is to provide a more comprehensive understanding of our approach, ensuring that the rationale behind our choices is transparent to the reader.
>
> We appreciate your insights and look forward to addressing these points in the revised manuscript. If you have any further questions or suggestions, please feel free to communicate them.
>
> [1] Le, C. P., Dong, J., Soltani, M., & Tarokh, V. (2021, October). Task Affinity with Maximum Bipartite Matching in Few-Shot Learning. In International Conference on Learning Representations.

---

### Official Review · Reviewer_DKw7 · 2023-11-01

**Soundness:** 2 fair
**Presentation:** 1 poor
**Contribution:** 1 poor
**Rating:** 3
**Confidence:** 4

**Summary:**

The paper suggests a generative learning method using a conditional generative adversarial network (cGAN) for the continual learning framework. They introduce a new score metric named Discriminator-based Mode Affinity Score, to measure the similarity of the target image class with source classes. This score is obtained by comparing the approximated Hessian matrix of the discriminator in cGAN frameworks on the loss of generated source images given the source image and the target image.

**Strengths:**

Suggest a new method for generative continual learning method and achieve improved performance against the baselines.
The suggested score metric seems to pick similar source classes with the target class.

**Weaknesses:**

- Overall, the motivation and presentation are weak. The need for continual learning for generative models and their own challenges beyond conventional continual learning scenarios and challenges are not well discussed, and this makes the reader feel that this work is a simple incremental work by transferring conditional generative adversarial networks on continual learning setting with simple repeats the well-known challenges - catastrophic forgetting - again.

- Limited investigation of 'modes'. The paper only assumes a mode is a class. However, this seems not realistic and outdated since recent generative models already have a surprising generalization and zero-shot ability on various styles/classes in a single model. Simple incremental learning in each 'class' means nothing these days.

- Similarly, tasks are too simple. Evaluation with MNIST/CIFAR and Flower dataset is a bit far from the recent generative model and/or continual learning trends. I recommend ImageNet/Coco + a, which can be better candidates. Additionally, baselines are also too old (most of them were published around three to four years ago). When we consider this venue for 2024, it is hard to confirm that the proposed idea and baselines are sufficiently strong compared to its counterparts/alternatives.

- No empirical comparison of the proposed metric. I fail to find the merits of the suggested score metric compared to other possible approaches to select similar source classes/modes with the target one, including FiD or other various types of metrics such as mutual information / KL or JS divergence metrics on their embeddings, etc. There are tons of techniques to meet the same purpose, but no comparison or demonstration to show the impact/strengths of the suggested score is provided.

**Questions:**

.

---

> ### Author Response · Authors · 2023-11-15
> **Response to Reviewer DKw7**
>
> Thank you for your feedback, and I apologize for any confusion. Below are the responses addressing your concerns:
> * Our research introduces a novel task affinity measure tailored specifically for generative tasks within conditional Generative Adversarial Networks (cGANs). Complementing this, we present a novel distance-based continual learning framework designed to refine cGAN models. The motivation driving our adoption of this distance-based approach stems from the imperative need to harness pre-existing well-trained models for swift adaptation to new tasks. For example, consider a model proficient in generating diverse types of cancer images. Our objective is to consistently enhance its training as new data becomes available, thereby ensuring its adaptability and ongoing improvement.
> * The replay mechanism functions as an auxiliary strategy to alleviate catastrophic forgetting, contributing to an overall enhancement in performance. We acknowledge that its inclusion is not a novel contribution to our work. Instead, it is motivated by the widespread application of similar mechanisms in various continual learning approaches, akin to techniques such as distillation. This underscores its common adoption in the field, emphasizing its recognized efficacy.
> * Our focus lies in addressing the challenge of generating images aligned with specific labels. Consequently, as new types of data are introduced, we aim to train the model on the images associated with the new class while preventing any loss of knowledge regarding existing classes. In the realm of continual learning for generative tasks, the overarching objective is to expand the model's knowledge base without experiencing forgetfulness. In our approach, the efficiency stems from the ability to seamlessly learn N new classes of data incrementally. This is achieved by capitalizing on the advantage of connecting the new data with the relevant previously learned classes. Our method demonstrates a notable advantage over other state-of-the-art approaches, particularly in terms of fine-tuning and training time.
> * In this study, we meticulously compare our approach with the latest state-of-the-art methods in the continually evolving field of continual learning for generative tasks, notwithstanding that some of these methods may date back 3-4 years. We are happy to incorporate any new references that may emerge in subsequent revisions.
> * It's crucial to highlight that our approach diverges from zero-shot or few-shot paradigms. In those contexts, the assumption typically revolves around the model being trained on biased yet relevant data. In our work, we contend with distinct image classes, and our objective is to discern the most akin classes for effective knowledge transfer.
> * In Section 4, our analysis encompasses a comparison with diverse methods, notably incorporating Sequential Fine-tuning (Zhai et al., 2019), which integrates the FID score for fine-tuning.

---

### Meta-Review · Area_Chair_WTrp · 2023-12-08

**Metareview:**

This paper proposes a novel method for continuous learning of class conditional GANs, based on replay of existing modes and matching new data to existing modes. Although (part of ) the reviewers find the studied problem relevant, they raise a number of concerns that are not fully addressed by the rebuttal. None of the reviewers recommends accepting the paper.

**Justification For Why Not Higher Score:**

None of the reviewers recommends accepting the paper.

**Justification For Why Not Lower Score:**

N/A

---

### Decision · Program_Chairs · 2024-01-16

Reject